# A Neural Compositional Paradigm
# for Image Captioning

**Bo Dai** [1]    **Sanja Fidler** [2,3,4]    **Dahua Lin** [1]

[1] CUHK-SenseTime Joint Lab, The Chinese University of Hong Kong
[2] University of Toronto    [3] Vector Institute    [4] NVIDIA

bdai@ie.cuhk.edu.hk    fidler@cs.toronto.edu    dhlin@ie.cuhk.edu.hk

## Abstract

Mainstream captioning models often follow a sequential structure to generate captions, leading to issues such as introduction of irrelevant semantics, lack of diversity in the generated captions, and inadequate generalization performance. In this paper, we present an alternative paradigm for image captioning, which factorizes the captioning procedure into two stages: (1) extracting an *explicit* semantic representation from the given image; and (2) constructing the caption based on a recursive *compositional* procedure in a bottom-up manner. Compared to conventional ones, our paradigm better preserves the semantic content through an explicit factorization of semantics and syntax. By using the compositional generation procedure, caption construction follows a recursive structure, which naturally fits the properties of human language. Moreover, the proposed compositional procedure requires less data to train, generalizes better, and yields more diverse captions.

## 1   Introduction

Image captioning, the task to generate short descriptions for given images, has received increasing attention in recent years. State-of-the-art models [1, 2, 3, 4] mostly adopt the encoder-decoder paradigm [3], where the content of the given image is first encoded via a convolutional network into a feature vector, which is then decoded into a caption via a recurrent network. In particular, the words in the caption are produced in a *sequential* manner – the choice of each word depends on both the preceding word and the image feature. Despite its simplicity and the effectiveness shown on various benchmarks [5, 6], the sequential model has a fundamental problem. Specifically, it could not reflect the *inherent* hierarchical structures of natural languages [7, 8] in image captioning and other generation tasks, although it could implicitly capture such structures in tasks taking the complete sentences as input, *e.g.* parsing [9], and classification [10].

As a result, sequential models have several significant drawbacks. First, they rely excessively on n-gram statistics rather than hierarchical dependencies among words in a caption. Second, such models usually favor the frequent n-grams [11] in the training set, which, as shown in Figure 1, may lead to captions that are only correct *syntactically* but not *semantically*, containing semantic concepts that are irrelevant to the conditioned image. Third, the entanglement of syntactic rules and semantics obscures the dependency structure and makes sequential models difficult to generalize.

To tackle these issues, we propose a new paradigm for image captioning, where the extraction of semantics (*i.e. what to say*) and the construction of syntactically correct captions (*i.e. how to say*) are decomposed into two stages. Specifically, it derives an *explicit* representation of the semantic content of the given image, which comprises a set of noun-phrases, *e.g. a white cat*, *a cloudy sky* or *two men*. With these noun-phrases as the basis, it then proceeds to construct the caption through *recursive composition* until a complete caption is obtained. In particular, at each step of the composition, a higher-level phrase is formed by joining two selected sub-phrases via a connecting phrase. It is

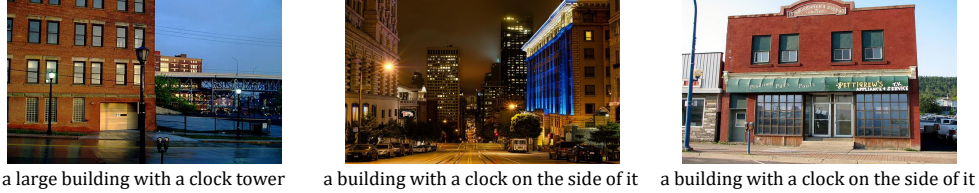

a large building with a clock tower     a building with a clock on the side of it     a building with a clock on the side of it

Figure 1: This figure shows three test images in MS-COCO [5] with captions generated by the neural image captioner [3], which contain n-gram *building with a clock* that appeared frequently in the training set but is not semantically correct for these images.

noteworthy that the compositional procedure described above is not a hand-crafted algorithm. Instead, it consists of two parametric modular nets, a *connecting module* for phrase composition and an *evaluation module* for deciding the completeness of phrases.

The proposed paradigm has several key advantages compared to conventional captioning models: (1) The factorization of *semantics* and *syntax* not only better preserves the semantic content of the given image but also makes caption generation easy to interpret and control. (2) The recursive composition procedure naturally reflects the inherent structures of natural language and allows the hierarchical dependencies among words and phrases to be captured. Through a series of ablative studies, we show that the proposed paradigm can effectively increase the diversity of the generated captions while preserving semantic correctness. It also generalizes better to new data and can maintain reasonably good performance when the number of available training data is small.

## 2 Related Work

Literature in image captioning is vast, with the increased interest received in the neural network era. The early approaches were bottom-up and detection based, where a set of visual concepts such as objects and attributes were extracted from images [12, 13]. These concepts were then assembled into captions by filling the blanks in pre-defined templates [13, 14], learned templates [15], or served as anchors to retrieve the most similar captions from the training set [16, 12].

Recent works on image captioning adopt an alternative paradigm, which applies convolutional neural networks [17] as image representation, followed by recurrent neural networks [18] for caption generation. Specifically, Vinyals *et al* [3] proposed the *neural image captioner*, which represents the input image with a single feature vector, and uses an LSTM [18] conditioned on this vector to generate words one by one. Xu *et al* [4] extended their work by representing the input image with a set of feature vectors, and applied an attention mechanism to these vectors at every time step of the recurrent decoder in order to extract the most relevant image information. Lu *et al* [1] adjusted the attention computation to also attend to the already generated text. Anderson *et al* [2] added an additional LSTM to better control the attention computation. Dai *et al* [19] reformulated the latent states as 2D maps to better capture the semantic information in the input image. Some of the recent approaches directly extract phrases or semantic words from the input image. Yao *et al* [20] predicted the occurrences of frequent training words, where the prediction is fed into the LSTM as an additional feature vector. Tan *et al* [21] treated noun-phrases as hyper-words and added them into the vocabulary, such that the decoder was able to produce a full noun-phrase in one time step instead of a single word. In [22], the authors proposed a hierarchical approach where one LSTM decides on the phrases to produce, while the second-level LSTM produced words for each phrase.

Despite the improvement over the model architectures, all these approaches generate captions *sequentially*. This tends to favor frequent n-grams [11, 23], leading to issues such as incorrect semantic coverage, and lack of diversity. On the contrary, our proposed paradigm proceeds in a bottom-up manner, by representing the input image with a set of noun-phrases, and then constructs captions according to a recursive composition procedure. With such explicit disentanglement between semantics and syntax, the recursive composition procedure preserves semantics more effectively, requires less data to learn, and also leads to more diverse captions.

Work conceptually related to ours is by Kuznetsova *et al* [24], which mines four types of phrases including noun-phrases from the training captions, and generates captions by selecting one phrase from each category and composes them via dynamic programming. Since the composition procedure is not recursive, it can only generate captions containing a single object, thus limiting the versatile

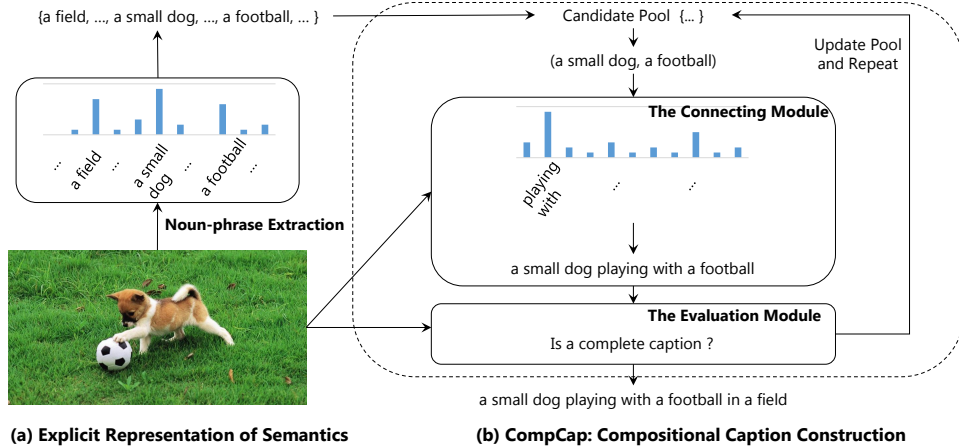

(a) Explicit Representation of Semantics      (b) CompCap: Compositional Caption Construction

Figure 2: An overview of the proposed compositional paradigm. A set of noun-phrases is extracted from the input image first, serving as the initial pool of phrases for the compositional generation procedure. The procedure then recursively uses a connecting module to compose two phrases from the pool into a longer phrase, until an evaluation module determines that a complete caption is obtained.

nature of image description. In our work, any number of phrases can be composed, and we exploit powerful neural networks to learn plausible compositions.

## 3 Compositional Captioning

The structure of natural language is inherently *hierarchical* [8, 7], where the typical parsing of a sentence takes the form of trees [25, 26, 27]. Hence, it's natural to produce captions following such a hierarchical structure. Specifically, we propose a two-stage framework for image captioning, as shown in Figure 2. Given an image, we first derive a set of noun-phrases as an explicit semantic representation. We then construct the caption in a bottom-up manner, via a recursive compositional procedure which we refer to as **CompCap**. This procedure can be considered as an *inverse* of the sentence parsing process. Unlike mainstream captioning models that primarily rely on the n-gram statistics among consecutive words, CompCap can take into account the nonsequential dependencies among words and phrases of a sentence. In what follows, we will present these two stages in more detail.

### 3.1 Explicit Representation of Semantics

Conventional captioning methods usually encode the content of the given image into feature vectors, which are often difficult to interpret. In our framework, we represent the image semantics *explicitly* by a set of *noun-phrases*, *e.g. "a black cat"*, *"a cloudy sky"* and *"two boys"*. These noun-phrases can capture not only the object categories but also the associated attributes.

Next, we briefly introduce how we extract such noun-phrases from the input image. It's worth noting that extracting such explicit representation for an image is essentially related to tasks of visual understanding. While more sophisticated techniques can be applied such as object detection [28] and attribute recognition [29], we present our approach here in order to complete the paradigm.

In our study, we found that the number of distinct noun-phrases in a dataset is significantly smaller than the number of images. For example, MS-COCO [5] contains $120K$ images but only about $3K$ distinct noun-phrases in the associated captions. Given this observation, it is reasonable to formalize the task of noun-phrase extraction as a multi-label classification problem.

Specifically, we derive a list of distinct noun-phrases $\{NP_1, NP_2, ..., NP_K\}$ from the training captions by parsing the captions and selecting those noun-phrases that occur for more than 50 times. We treat each selected noun-phrase as a *class*. Given an image $I$, we first extract the visual feature $\mathbf{v}$ via a Convolutional Neural Network as $\mathbf{v} = \text{CNN}(I)$, and further encode it via two fully-connected layers as $\mathbf{x} = F(\mathbf{v})$. We then perform binary classification for each noun-phrase

$NP_k$ as $S_C(NP_k|I) = \sigma(\mathbf{w}_k^T\mathbf{x})$, where $\mathbf{w}_k$ is the weight vector corresponding to the class $NP_k$ and $\sigma$ denotes the sigmoid function.

Given $\{S_C(NP_k|I)\}_k$, the scores for individual noun-phrases, we choose to represent the input image using $n$ of them with top scores. While the selected noun-phrases may contain semantically similar concepts, we further prune this set through *Semantic Non-Maximum Suppression*, where only those noun-phrases whose scores are the maximum among similar phrases are retained.

### 3.2 Recursive Composition of Captions

Starting with a set of noun-phrases, we construct the caption through a recursive compositional procedure called **CompCap**. We first provide an overview, and describe details of all the components in the following paragraphs.

At each step, CompCap maintains a phrase pool $\mathcal{P}$, and scans all *ordered* pairs of phrases from $\mathcal{P}$. For each ordered pair $P^{(l)}$ and $P^{(r)}$, a *Connecting Module (C-Module)* is applied to generate a sequence of words, denoted as $P^{(m)}$, to connect the two phrases in a plausible way. This yields a longer phrase in the form of $P^{(l)} \oplus P^{(m)} \oplus P^{(r)}$, where $\oplus$ denotes the operation of sequence concatenation. The C-Module also computes a score for $P^{(l)} \oplus P^{(m)} \oplus P^{(r)}$. Among all phrases that can be composed from scanned pairs, we choose the one with the maximum connecting score as the new phrase $P_{\text{new}}$. A parametric module could also be used to determine $P_{new}$.

Subsequently, we apply an *Evaluation Module (E-Module)* to assess whether $P_{new}$ is a *complete* caption. If $P_{new}$ is determined to be complete, we take it as the resulting caption; otherwise, we update the pool $\mathcal{P}$ by replacing the corresponding constituents $P^{(l)}$ and $P^{(r)}$ with $P_{new}$, and invoke the pair selection and connection process again based on the updated pool. The procedure continues until a complete caption is obtained or only a single phrase remains in $\mathcal{P}$.

We next introduce the connecting and the evaluation module, respectively.

**The Connecting Module.** The *Connecting Module (C-Module)* aims to select a *connecting phrase* $P^{(m)}$ given both the left and right phrases $P^{(l)}$ and $P^{(r)}$, and to evaluate the *connecting score* $S(P^{(m)} \mid P^{(l)}, P^{(r)}, I)$. While this task is closely related to the task of filling in the blanks of captions [30], we empirically found that the conventional way of using an LSTM to decode the intermediate words fails. One possible reason is that inputs in [30] are always prefix and suffix of a complete caption. The C-Module, by contrast, mainly deals with incomplete ones, constituting a significantly larger space. In this work, we adopt an alternative strategy, namely, to treat the generation of connecting phrases as a classification problem. This is motivated by the observation that the number of distinct connecting phrases is actually limited in the proposed paradigm, since semantic words such as nouns and adjectives are not involved in the connecting phrases. For example, in MS-COCO [5], there are over 1 million samples collected for the connecting module, which contain only about $1,000$ distinct connecting phrases.

Specifically, we mine a set of distinct connecting sequences from the training captions, denoted as $\{P_1^{(m)}, \ldots, P_L^{(m)}\}$, and treat them as different classes. This can be done by walking along the parsing trees of captions. We then define the connecting module as a classifier, which takes the left and right phrases $P^{(l)}$ and $P^{(r)}$ as input and outputs a normalized score $S(P_j^{(m)} \mid P^{(l)}, P^{(r)}, I)$ for each $j \in \{1, \ldots, L\}$.

In particular, we adopt a two-level LSTM model [2] to encode $P^{(l)}$ and $P^{(r)}$ respectively, as shown in Figure 3. Here, $\mathbf{x}_t$ is the word embedding for $t$-th word, and $\mathbf{v}$ and $\{\mathbf{u}_1, ..., \mathbf{u}_M\}$ are, respectively, global and regional image features extracted from a Convolutional Neural Network. In this model, the low-level LSTM controls the attention while interacting with the visual features, and the high-level LSTM drives the evolution of the encoded state. The encoders for $P^{(l)}$ and $P^{(r)}$ share the same structure but have different parameters, as one phrase should be encoded differently based on its place in the ordered pair. Their encodings, denoted by $\mathbf{z}^{(l)}$ and $\mathbf{z}^{(r)}$, go through two fully-connected layers followed by a softmax layer, as

$$S(P_j^{(m)} \mid P^{(l)}, P^{(r)}, I) = \text{Softmax}(\mathbf{W}_{combine} \cdot (\mathbf{W}_l \cdot \mathbf{z}^{(l)} + \mathbf{W}_r \cdot \mathbf{z}^{(r)}))|_j, \quad \forall \ j = 1, ..., L. \quad (1)$$

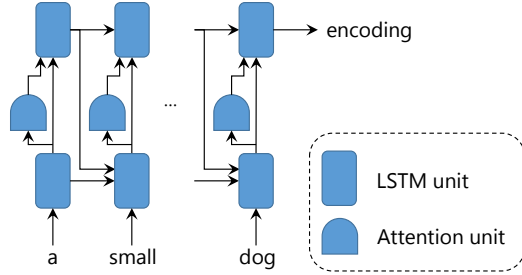

$$\mathbf{h}_0^{(att)} = \mathbf{h}_0^{(lan)} = \mathbf{0}$$

$$\mathbf{h}_t^{(att)} = \text{LSTM}(\mathbf{x}_t, \mathbf{v}, \mathbf{h}_{t-1}^{(lan)}, \mathbf{h}_{t-1}^{(att)})$$

$$\mathbf{a}_t = \text{Attention}(\mathbf{h}_t^{(att)}, \mathbf{u}_1, ..., \mathbf{u}_M)$$

$$\mathbf{h}_t^{(lan)} = \text{LSTM}(\mathbf{a}_t, \mathbf{h}_t^{(att)}, \mathbf{h}_{t-1}^{(lan)})$$

$$\mathbf{z} = \mathbf{h}_T^{(lan)}$$

(a) Structure of the Phrase Encoder　　　　(b) Computation of the Phrase Encoder

Figure 3: This figure shows the two-level LSTM used to encode phrases in the connecting and evaluation modules. **Left:** the structure of the phrase encoder, **right:** its updating formulas.

The values of the softmax output, *i.e.* $S(P_j^{(m)} \mid P^{(l)}, P^{(r)}, I)$, are then used as the *connecting scores*, and the connecting phrase that yields the highest connecting score is chosen to connect $P^{(l)}$ and $P^{(r)}$.

While not all pairs of $P^{(l)}$ and $P^{(r)}$ can be connected into a longer phrase, in practice a virtual connecting phrase $P_{\text{neg}}^{(m)}$ is added to serve as a negative class.

Based on the C-Module, we compute the score for a phrase as follow. For each noun-phrase $P$ in the initial set, we set its score to be the binary classification score $S_C(P|I)$ obtained in the phrase-from-image stage. For each longer phrase produced via the C-Module, its score is computed as

$$S\left(P^{(l)} \oplus P^{(m)} \oplus P^{(r)} \mid I\right) = S\left(P^{(l)} \mid I\right) + S\left(P^{(r)} \mid I\right) + S\left(P^{(m)} \mid P^{(l)}, P^{(r)}, I\right). \quad (2)$$

**The Evaluation Module.** The *Evaluation Module (E-Module)* is used to determine whether a phrase is a complete caption. Specifically, given an input phrase $P$, the E-Module encodes it into a vector $\mathbf{z}_e$, using a two-level LSTM model as described above, and then evaluates the probability of $P$ being a complete caption as

$$\Pr(P \text{ is complete}) = \sigma\left(\mathbf{w}_{cp}^T \mathbf{z}_e\right). \quad (3)$$

It's worth noting that other properties could also be checked by the E-Module besides the completeness. *e.g.* using a caption evaluator [11] to check the quality of captions.

**Extensions.** Instead of following the greedy search strategy described above, we can extend the framework for generating diverse captions for a given image, via beam search or probabilistic sampling. Particularly, we can retain multiple ordered pairs at each step and multiple connecting sequences for each retained pair. In this way, we can form multiple beams for beam search, and thus avoid being stuck in local minima. Another possibility is to generate diverse captions via probabilistic sampling, *e.g.* sampling a part of the ordered pairs for pair selection instead of using all of them, or sampling the connecting sequences based on their normalized scores instead of choosing the one that yields the highest score.

The framework can also be extended to incorporate user preferences or other conditions, as it consists of operations that are interpretable and controllable. For example, one can influence the resultant captions by filtering the initial noun phrases or modulating their scores. Such control is much easier to implement on an explicit representation, *i.e.* a set of noun phrases, than on an encoded feature vector. We show examples in the Experimental section.

## 4 Experiments

### 4.1 Experiment Settings

All experiments are conducted on MS-COCO [5] and Flickr30k [6]. There are $123,287$ images and $31,783$ images respectively in MS-COCO and Flickr30k, each of which has $5$ ground-truth captions.

Table 1: This table lists results of different methods on MS-COCO [5] and Flickr30k [6]. Results of CompCap using ground-truth noun-phrases and composing orders are shown in the last two rows.

| | COCO-offline | | | | | Flickr30k | | | | |
|---|---|---|---|---|---|---|---|---|---|---|
| | SP | CD | B4 | RG | MT | SP | CD | B4 | RG | MT |
| NIC [3] | 17.4 | 92.6 | 30.2 | 52.3 | 24.3 | 12.0 | 40.7 | 19.9 | 42.9 | 18.0 |
| AdapAtt [1] | 18.1 | 97.0 | 31.2 | 53.0 | 25.0 | 13.4 | 48.2 | 23.3 | 45.5 | 19.3 |
| TopDown [2] | 18.7 | **101.1** | **32.4** | **53.8** | **25.7** | 13.8 | **49.8** | **23.7** | **45.6** | **19.7** |
| LSTM-A5 [20] | 18.0 | 96.6 | 31.2 | 53.0 | 24.9 | 12.2 | 43.7 | 20.4 | 43.8 | 18.2 |
| CompCap + $\text{Pred}_{np}$ | **19.9** | 86.2 | 25.1 | 47.8 | 24.3 | **14.9** | 42.0 | 16.4 | 39.4 | 19.0 |
| CompCap + $\text{GT}_{np}$ | 36.8 | 122.2 | 42.8 | 55.3 | 33.6 | 31.9 | 89.7 | 37.8 | 50.5 | 28.7 |
| CompCap + $\text{GT}_{np}$ + $\text{GT}_{order}$ | 33.8 | 182.6 | 64.1 | 82.4 | 45.1 | 29.8 | 132.8 | 54.9 | 77.1 | 39.6 |

We follow the splits in [31] for both datasets. In both datasets, the vocabulary is obtained by turning words to lowercase and removing words that have non-alphabet characters and appear less than 5 times. The removed words are replaced with a special token *UNK*, resulting in a vocabulary of size $9,487$ for MS-COCO, and $7,000$ for Flickr30k. In addition, training captions are truncated to have at most $18$ words. To collect training data for the connecting module and the evaluation module, we further parse ground-truth captions into trees using NLPtookit [32].

In all experiments, C-Module and E-Module are separately trained as in two standard classification tasks. Consequently, the recursive compositional procedure is modularized, making it less sensitive to training statistics in terms of the composing order, and generalizes better. When testing, each step of the procedure is done via two forward passes (one for each module). We empirically found that a complete caption generally requires 2 or 3 steps to obtain.

Several representative methods are compared with CompCap. They are 1) *Neural Image Captioner (NIC)* [3], which is the backbone network for state-of-the-art captioning models. 2) *AdapAtt* [1] and 3) *TopDown* [2] are methods that apply the attention mechanism and obtain state-of-the-art performances. While all of these baselines encode images as semantical feature vectors, we also compare CompCap with 4) *LSTM-A5* [20], which predicts the occurrence of semantical concepts as additional visual features. Subsequently, besides being used to extract noun-phrases that fed into CompCap, predictions of the noun-phrase classifiers also serve as additional features for *LSTM-A5*.

To ensure a fair comparsion, we have re-implemented all methods, and train all methods using the same hyperparameters. Specifically, we use ResNet-152 [17] pretrained on ImageNet [33] to extract image features, where activations of the last convolutional and fully-connected layer are used respectively as the regional and global feature vectors. During training, we fix ResNet-152 without finetuning, and set the learning rate to be $0.0001$ for all methods. When testing, for all methods we select parameters that obtain best performance on the validation set to generate captions. Beam-search of size $3$ is used for baselines. As for CompCap, we empirically select $n = 7$ noun-phrases with top scores to represent the input image, which is a trade-off between semantics and syntax, as shown in Figure 8. Beam-search of size $3$ is used for pair selection, while no beam-search is used for connecting phrase selection.

## 4.2 Experiment Results

**General Comparison.** We compare the quality of the generated captions on the offline test set of MS-COCO and the test set of Flickr30k, in terms of SPICE (SP) [34], CIDEr (CD) [35], BLEU-4 (B4) [36], ROUGE (RG) [37], and METEOR (MT) [38]. As shown in Table 1, among all methods, CompCap with predicted noun-phrases obtains the best results under the SPICE metric, which has higher correlation with human judgements [34], but is inferior to baselines in terms of CIDEr, BLEU-4, ROUGE and METEOR. These results well reflect the properties of methods that generate captions sequentially and compositionally. Specifically, while SPICE focuses on semantical analysis, metrics including CIDEr, BLEU-4, ROUGE and METEOR are known to favor frequent training n-grams [11], which are more likely to appear when following a sequential generation procedure. On the contrary, the compositional generation procedure preserves semantic content more effectively, but may contain more n-grams that are not observed in the training set.

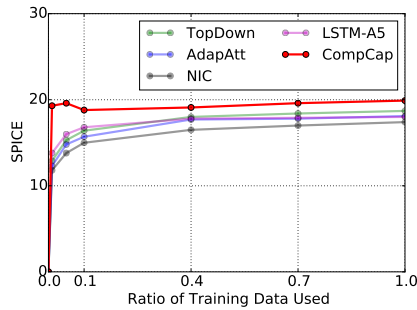

(a) SPICE: Train on COCO using less data

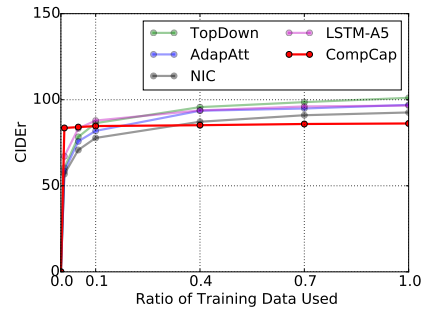

(b) CIDEr: Train on COCO using less data

Figure 4: This figure shows the performance curves of different methods when less data is used for training. Unlike baselines, CompCap obtains stable results as the ratio of used data decreases.

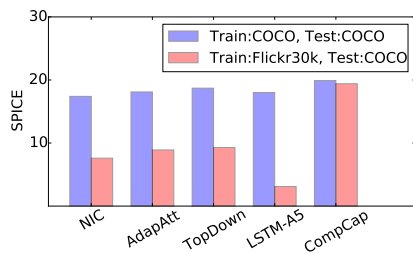

(a) SPICE: COCO -> Flickr30k

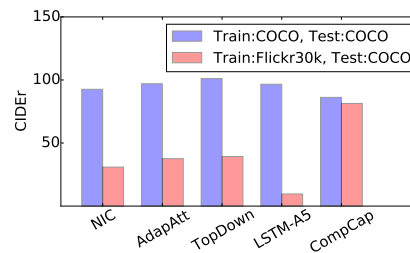

(b) CIDEr: COCO -> Flickr30k

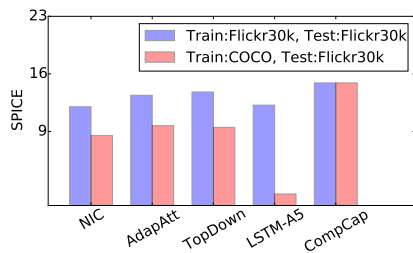

(c) SPICE: Flickr30k -> COCO

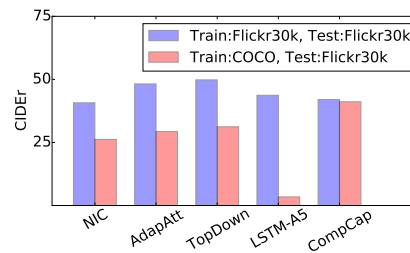

(d) CIDEr: Flickr30k -> COCO

Figure 5: This figure compares the generalization ability of different methods, where they are trained on one dataset, and tested on the other. Compared to baselines, CompCap is shown to generalize better across datasets.

An ablation study is also conducted on components of the proposed compositional paradigm, as shown in the last three rows of Table 1. In particular, we represented the input image with ground-truth noun-phrases collected from 5 associated captions, leading to a significant boost in terms of all metrics. This indicates that CompCap effectively preserves the semantic content, and the better the semantic understanding we have for the input image, CompCap is able to generate better captions for us. Moreover, we also randomly picked one ground-truth caption, and followed its composing order to integrate its noun-phrases into a complete caption, so that CompCap only accounts for connecting phrase selection. As a result, metrics except for SPICE obtain further boost, which is reasonable as we only use a part of all ground-truth noun-phrases, and frequent training n-grams are more likely to appear following some ground-truth composing order.

**Generalization Analysis.** As the proposed compositional paradigm disentangles semantics and syntax into two stages, and CompCap mainly accounts for composing semantics into a syntactically correct caption, CompCap is good at handling out-of-domain semantic content, and requires less data to learn. To verify this hypothesis, we conducted two studies. In the first experiment, we controlled the ratio of data used to train the baselines and modules of CompCap, while leaving the noun-phrase classifiers being trained on full data. The resulting curves in terms of SPICE and CIDEr are shown in

Table 2: This table measures the diversity of generated captions from various aspects, which suggests CompCap is able to generate more diverse captions.

|  | | NIC [3] | AdapAtt [1] | TopDown [2] | LSTM-A5 [20] | CompCap |
|---|---|---|---|---|---|---|
| COCO | Novel Caption Ratio | 44.53% | 49.34% | 45.05% | 50.06% | **90.48%** |
| | Unique Caption Ratio | 55.05% | 59.14% | 61.58% | 62.61% | **83.86%** |
| | Diversity (Dataset) | 7.69 | 7.86 | 7.99 | 7.77 | **9.85** |
| | Diversity (Image) | 2.25 | 3.61 | 2.30 | 3.70 | **5.57** |
| | Vocabulary Usage | 6.75% | 7.22% | 7.97% | 8.14% | **9.18%** |

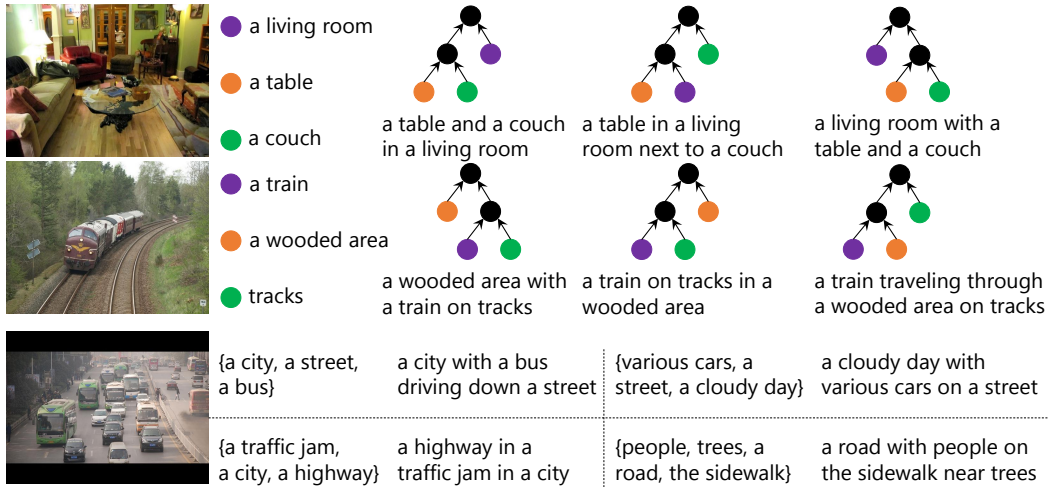

Figure 6: This figure shows images with diverse captions generated by CompCap. In first two rows, captions are generated with same noun-phrases but different composing orders. And in the last row, captions are generated with different sets of noun-phrases.

Figure 4, while other metrics follow similar trends. Compared to baselines, CompCap is steady and learns how to compose captions even only 1% of the data is used.

In the second study, we trained baselines and CompCap on MS-COCO/Flickr30k, and tested them on Flickr30k/MS-COCO. Again, the noun-phrase classifiers are trained with in-domain data. The results in terms of SPICE and CIDEr are shown in Figure 5, where significant drops are observed for the baselines. On the contrary, competitive results are obtained for CompCap trained using in-domain and out-of-domain data, which suggests the benefit of disentangling semantics and syntax, as the distribution of semantics often varies from dataset to dataset, but the distribution of syntax is relatively stable across datasets.

**Diversity Analysis.** One important property of CompCap is the ability to generate diverse captions, as these can be obtained by varying the involved noun-phrases or the composing order. To analyze the diversity of captions, we computed five metrics that evaluate the degree of diversity from various aspects. As shown in Table 2, we computed the ratio of novel captions and unique captions [39], which respectively account for the percentage of captions that are not observed in the training set, and the percentage of distinct captions among all generated captions. We further computed the percentage of words in the vocabulary that are used to generate captions, referred to as the vocabulary usage.

Finally, we quantify the diversity of a set of captions by averaging their pair-wise editing distances, which leads to two additional metrics. Specifically, when only a *single* caption is generated for each image, we report the average distance over captions of different images, which is defined as the diversity at the dataset level. If *multiple* captions are generated for each image, we then compute the average distance over captions of the same image, followed by another average over all images. The final average is reported as the diversity at the image level. The former measures how diverse the captions are for different images, and the latter measures how diverse the captions are for a single image. In practice, we use 5 captions with top scores in the beam search to compute the diversity at the image level, for each method.

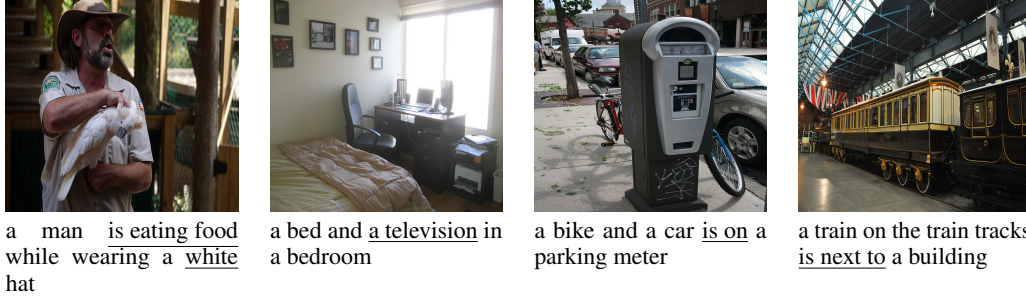

| a man is eating food while wearing a white hat | a bed and a television in a bedroom | a bike and a car is on a parking meter | a train on the train tracks is next to a building |

Figure 7: Some failure cases are included in this figure, where errors are highlighted by underlines. The first two cases are related to errors in the first stage (*i.e.* semantic extraction), and the last two cases are related to the second stage (*i.e.* caption construction).

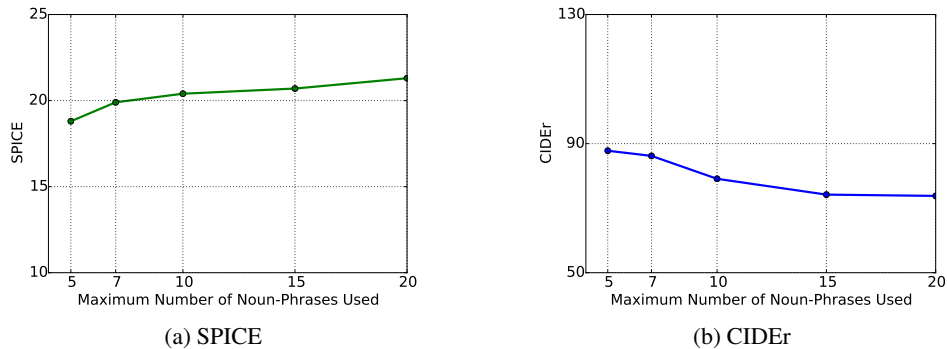

(a) SPICE

(b) CIDEr

Figure 8: As shown in this figure, as the maximum number of noun-phrases increases, SPICE improves but CIDEr decreases, which indicates although introducing more noun-phrases could lead to semantically richer captions, it may risk the syntactic correctness.

CompCap obtained the best results in all metrics, which suggests that captions generated by CompCap are diverse and novel. We further show qualitative samples in Figure 6, where captions are generated following different composing orders, or using different noun-phrases.

**Error Analysis.** We include several failure cases in Figure 7, which share similar errors with the results listed in Figure 1. However, the causes are fundamentally different. Generally, errors in captions generated by CompCap mainly come from the misunderstanding of the input visual content, which could be fixed by applying more sophisticated techniques in the stage of noun-phrase extraction. It's, by contrast, an intrinsic property for sequential models to favor frequent n-grams. With a perfect understanding of the visual content, sequential models may still generate captions containing incorrect frequent n-grams.

# 5   Conclusion

In this paper, we propose a novel paradigm for image captioning. While the typical existing approaches encode images using feature vectors and generate captions sequentially, the proposed method generates captions in a compositional manner. In particular, our approach factorizes the captioning procedure into two stages. In the first stage, an explicit representation of the input image, consisting of noun-phrases, is extracted. In the second stage, a recursive compositional procedure is applied to assemble extracted noun-phrases into a caption. As a result, caption generation follows a hierarchical structure, which naturally fits the properties of human language. On two datasets, the proposed compositional procedure is shown to preserve semantics more effectively, require less data to train, generalize better across datasets, and yield more diverse captions.

**Acknowledgement**   This work is partially supported by the Big Data Collaboration Research grant from SenseTime Group (CUHK Agreement No. TS1610626), the General Research Fund (GRF) of Hong Kong (No. 14236516).

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
