[Reviews · NeurIPS 2018]

Reviewer 1



The paper proposes a compositional method to generate image captions. First, a classifier is trained to generate a list of none-phrases for an image. Then two none-phrases are connected by a connecting sequence to form a longer phrase. The algorithm stops when a long phrase is evaluated to be a complete caption. The proposed method achieves comparable SPICE score with state-of-the-art methods. Pros: 1. The compositional paradigm is able to model the hierarchical structure of natural language. 2. The compositional procedure consists of parametric modular nets, instead of hand-crafted procedures. Cons: 1. The authors claim that the sequential model cannot reflect the hierarchical structure of natural languages. However, sequential models are used to encode the phrases and determine whether a phrase is a complete caption. If a sequential model is unable to model the hierarchical structure of languages, how can it know whether a compositional phrase is a complete caption? 2. In Eq. (1), the left phrase and right phrase representation are added. How to determine the order of two phrases in the longer phrase. For example, how to decide it is "a dog playing a football" instead of "a football playing a dog". 3. In the second study of the generalization analysis, the none-phrase classifiers are trained with in-domain data. Are in-domain data used by other methods? 4. As building and clock occur frequently in the training set, is it possible the none-phase classifier misclassify a building to a clock? If so, will the proposed method have the same error illustrated in Figure 1?

Reviewer 2



*Summary* The paper proposes a novel approach to image captioning, which adopts a recursive approach to assembling concepts detected from a visual pipeline, based on picking noun phrases and connecting them with relation phrases, which disentangles the content from the syntacic information (to a first order). The paper then shows that using such an explicit factorization in the generation mecahnism can help achieve better diversity in captions, and better generalization with lesser amount of data, and really good results on the SPICE metric which focusses more on the syntactic structure of image cpations. *Strengths* + The paper has a pretty comprehensive list of references + The experimental results seem really thorough and consistently done, compared to most image captioning papers. The same CNN is used for all the methods and hyperparameters are all set very systematically. + It is very interesting that the proposed approach does really well on SPICE, but significantly poorly on other metrics, and it makes a lot of sense given the SPICE metric that such a method should do well on it, as well. + The experiment on using the ground truth noun phrases and using the compositional part of the model is also really nice, and gives an insight into the what the model has learnt and how performance with this paradigm might improve with better visual detectors. *Weakness* 1. Related work which would be good to cite is [A], which also first predicts the noun phrases and then uses that to influence the structure of the captionr, and It would be good to compare and contrast both conceptually what the differences are and otherwise. 2. L133-134: Not clear what we mean by not only the complete captions but also parts thereof, and generally argument around why filling in the blanks is hard is unconvincing here. Note that there are ways to do bi-directional beam search for fill in the blanks kind of applications [C]. 3. How time consuming is the forward pass to create the caption? It seems like a pretty computationally intensive approach. How sensitive is performance to the number of noun phrases which get selected? (L202). 4. How do the statistics of the relation phrases look like? How often are the rarer relation phrases actually predicted in a sentence? Chunking things into relation phrases could have the disadvantage that one might see individual words but not the phrase at once and now we are stuck with the problem of heavy tail in relations (which is more severe than heavy tail in words). 5. Fig. 4: Why is there a bump between 0.0 and 0.1 for CompCap, and why is performance so high even at 0.0? It is realy confusing. 6. What is the training objective used for training the whole model? How are modules and losses set up? Are all the operations performed differentiable? It seems like the model makes hard decisions when picking which noun phrase to use or which relation phrase to use, does this mean there are only sequence level rewards and the whole model is trained via REINFORCE/ policy gradient? It seems like the paper is incomplete in this regard. (*) 7. L227: in a lot of ways the experiment in this paragraph is not fair, since the baseline approaches are learning to caption from scratch while the proposed approach is using noun phrase detectors trained on the full data. It would be good to report what happens if we also train the noun phrase detectors only on a part of the data. (*) *Preliminary Evaluation* There is a major detail about the training objective and algorithms missing from the paper, however, the crux of the approach that is presented is very interesting, and the results as well as experimental analysis look great and are very thorough and creative. This would be an interesting contribution to the conference, but important details about the paper are missing, which should be fixed, marked with (*) above. References: [A]: Gan, Zhe, Chuang Gan, Xiaodong He, Yunchen Pu, Kenneth Tran, Jianfeng Gao, Lawrence Carin, and Li Deng. 2016. “Semantic Compositional Networks for Visual Captioning.” arXiv [cs.CV]. arXiv. http://arxiv.org/abs/1611.08002. [B]: Huang, Qiuyuan, Paul Smolensky, Xiaodong He, Li Deng, and Dapeng Wu. 2017. “Tensor Product Generation Networks.” arXiv [cs.CV]. arXiv. http://arxiv.org/abs/1709.09118. [C]: Sun, Qing, Stefan Lee, and Dhruv Batra. 2017. “Bidirectional Beam Search: Forward-Backward Inference in Neural Sequence Models for Fill-in-the-Blank Image Captioning.” arXiv [cs.CV]. arXiv. http://arxiv.org/abs/1705.08759.

Reviewer 3



The paper proposes an alternative architecture to mainstream captioning models which usually follow a sequential structure. The model has two stages: 1) generating explicit semantic representations given images using noun phrases. 2) composing the noun phrases using a recursive architecture to generate the final caption. - Idea/approach wise I think this is an interesting direction to go for generating captions, especially with the issues currently associated with captioning models (e.g. lack of diversity). - The paper is very well written. It is easy to follow the ideas and get to the details of the approach, the illustrations (figures) are motivating and clear. - While the current approach only outperforms the state of the art with one metric (SPICE), the experimental results are strong -- with good analysis to show where further improvements can be made (e.g. providing ground truth noun phrases), and where the strengths of the model is (generalization to other datasets). Things to improve: - It would be great to show the training and testing time of the model. For example, how long does it take to train the connecting module? Comparing all pairs of left and right might take a long time. - While SPICE is shown to have correlated more with human judgement, it would still be good to conduct some human evaluation experiments to strengthen the paper. - It would be nice to show some failure cases of the model, and see whether it is n-gram grammatical structure that fails in the most cases. - Are the modules trained jointly? If not, would joint training help?